# The Comparison of Perfectionism and Commitment between Professional and Amateur Golfers and the Association between Perfectionism and Commitment in the Two Groups

**DOI:** 10.3390/ijerph17165657

**Published:** 2020-08-05

**Authors:** Jae Jun Nam, Doug Hyun Han

**Affiliations:** 1Department of Golf, Korea Golf University, Hoeng Seong, Gang won-do 25247, Korea; 01197097765@nate.com; 2Department of Psychiatry, Chung Ang University Hospital, Heuckseok Ro 102, Dong Jack Gu, Seoul 06973, Korea

**Keywords:** golf, perfectionism traits, commitment, perfectionism

## Abstract

This study investigated differences in perfectionist traits and commitment between professional and amateur golfers, as well as correlations among perfectionist traits, commitment, and golf handicap. Using simple random sampling, 486 professional golfers (mean age = 22.1 ± 3.0, 52.1% female) and 233 amateur golfers (mean age = 44.8 ± 10.2, 55.8% female) were recruited and assessed using the Multidimensional Perfectionism Scale (MPS) and Expansion of Sports Commitment Model (ESCM). An ANCOVA, controlling for age, golf career length, and training time, revealed lower MPS self-oriented scores (10.3%; *F* = 8.9, *p* < 0.01; effect size [ES] = 0.498) and higher ESCM-Cognition (12.6%; *F* = 9.4, *p* < 0.01; ES = 0.691) and ESCM-Behavior (9.4%; *F* = 4.6, *p* = 0.03; ES = 0.479) scores in professional golfers than in amateur golfers. In partial correlations controlling for age, golf career length, and training time, professional golfers’ MPS scores were negatively associated with ESCM-Cognition scores (*r* = -0.30, *p* < 0.001). Professional golfers’ mean golf handicap was positively correlated with MPS total (*r* = 0.33, *p* < 0.01). Altogether, golfers seeking to attain high levels of performance must consider the mental aspect of golfing and find ways to maximize commitment levels while minimizing perfectionist traits.

## 1. Introduction

Perfectionism is a personality trait characterized by striving toward flawlessness and setting excessively high standards for performance [1]. There have been several studies about perfectionism in Korean athletes [2,3]. Yoon and Kim [2] reported that the traits of perfectionism in man-to-man sports were higher than those in record genre sports. Bum et al. [3] reported that self-oriented perfectionism and socially prescribed perfectionism would aggravate the level of stress and emotional exhaustion. In addition, perfectionism can arise from coach and parental pressure, which can disrupt cognitive, emotional, and behavioral aspects of sport performance [4]. In our previous studies of women professional golfers, perfectionism was associated with burnout and golf handicap [5]. Magnusson et al. suggested that perfectionism was associated with mental fatigue related to maladaptive coping strategies [6].

Perfectionism is also accompanied by tendencies to be overly critical of one’s behavior [7]. Perfectionist traits have been associated with obsessive traits [8], which are defined by recurrent and persistent thoughts, impulses, or images that are perceived as intrusive and inappropriate and can cause marked anxiety and distress [9]. In a study of 11 female patients and 11 matched healthy controls, van der Wee et al. [10] reported that obsessive traits would lead to increased deficits in response to higher levels of task difficulty in a working memory task. Interestingly, improved working memory capacity seems to correlate with an athletes’ highest level of skill [11,12,13,14,15,16,17]. Numerous studies have suggested that improved working memory is highly associated with processing speed [11,12,13]. Slow processing speed on the other hand, has been related to increased rates of inattention [14,15,16,17]. These results seem to be consistent with those reported by Hancock et al. [18], who suggested that faster processing speeds are positively correlated with high levels of commitment. However, there is yet to be any research linking perfectionism and commitment in sports or exercise.

Sport commitment represents a psychological state reflecting an athlete’s desire and resolve to continue his or her sport participation [19], and there is empirical support for the view that high levels of sport commitment accompany greater behavioral persistence [20]. Numerous studies have suggested that commitment is associated with high-level performance or peak experience in sports [21]. Swann et al. [22] reported that commitment in elite golfers, including pre-shot routines, psychological interventions, and a healthy physical state, were important factors for good performance. Similar to perfectionism, sport commitment was associated with the attitude of coaches and parental pressure [23,24], and can disrupt cognitive, emotional, and behavioral aspects of sport performance [25]. Taken together, mental fatigue induced by perfectionism in golfers may impair performance and sport commitment. With such research in mind, this study aimed to investigate the differences in perfectionist traits and commitment between professional and amateur golfers who were members of the Korean Golf Association (KGA). In addition, we investigated whether the level of perfectionist traits in professional golfers was related to commitment and performance, usually represented by average golf scores (i.e., golf handicap).

We hypothesized that perfectionist traits would negatively correlate with levels of commitment, but would positively correlate with golf performance, as measured by average scores based on golf handicap. In summary, we expected “low perfectionist traits” to be associated with excellent performance (i.e., lower golf handicap) during competition.

## 2. Methods

### 2.1. Design and Subjects

This study was designed as a cross-sectional case control study with two groups. All participants were recruited using simple random sampling. Participants included 486 professional golf athletes (mean age ± standard deviation = 22.1 ± 3.0 years) and 233 amateur golf athletes (44.8 ± 10.2 years) who were members of the KGA.

There were 233 (47.9%) and 253 (52.1%) male and female professional golfers, respectively, and 103 (44.2%) and 130 (55.8%) amateur golfers, respectively. The average golf career length of professional golfers and amateur golfers was 8.1 ± 3.5 and 12.6 ± 10.2 years, respectively. The average training time of professional and amateur golfers was 6.5 ± 2.6 and 3.2 ± 2.5 h per day, respectively. The average score of golf games (golf handicap) of professional golfers and amateur golfers was 73.7 ± 5.8 and 84.0 ± 11.1, respectively.

Participants in the professional group met the following inclusion criteria: (i) Division 1 golf players in the KGA; (ii) aged from 18 to 60 years; and (iii) more than 2 years of experience in the Korea Professional Golfers Association (KPGA) or Korea Ladies Professional Golfers Association (KLPGA). Participants in the amateur group met the following inclusion criteria: (i) participated in at least one amateur tournament per year in Korea in the previous 2 years; (ii) aged from 18 to 60 years; and (iii) no history of KPGA or KLPGA membership (Table 1).

All athletes were assessed with the Multidimensional Perfectionism Scale (MPS) and the Expansion of Sports Commitment Model (ESCM). Data from 18 professional athletes and 11 amateur athletes were excluded due to incomplete answers or large amounts of missing data. Participants gave written, informed consent to participate in the study. All data were collected on the golf courses of the above-mentioned tournament sites, with full cooperation of the KGA. Specifically, after the players arrived at the club house, the survey was conducted before they entered the locker room. The protocol of this study was approved by the Chung Ang University Hospital Review Board (C2014111).

### 2.2. Measurements

#### 2.2.1. Multidimensional Perfectionism Scale (MPS)-Korean

The 45-item Multidimensional Perfectionism Scale (MPS) was developed by Hewitt and Flett [26]. A Korean version of the MPS has been verified in Korean university students [27] with a reliability of α = 0.84 (self-oriented), α = 0.73 (other-oriented), and α = 0.76 (socially prescribed). In a confirmatory factor analysis, the model fit the data: χ^2^ = 103.98, *p* < 0.0001, goodness of fit index (GFI) = 0.93, normed fit index (NFI) = 0.90, Tucker–Lewis index (TLI) = 0.92, comparative fit index (CFI) = 0.94, standardized root mean square residual (SRMR) = 0.043, and root mean square error of approximation (RMSEA) (90% CI) = 0.08 [28].

This 45-question scale has three dimensions: self-oriented (e.g., “When I am working on something, I cannot relax until it is perfect”), other-oriented (e.g., “I am not likely to criticize someone for giving up too easily”), and socially prescribed perfectionism (e.g., “I find it difficult to meet others’ expectations of me”). Each dimension has 15 questions arranged on a seven-point Likert scale ranging from 1 (completely disagree) to 7 (completely agree). Higher total scores represent a higher tendency of perfectionism.

#### 2.2.2. Expansion of Sports Commitment Model (ESCM)-Korean

The 13-item Expansion of Sports Commitment Model (ESCM) was developed by Scanlan et al. [29]. The ESCM-Korean version has been verified in Korean sports participants with reliability scores of α = 0.86 (ESCM-cognitive commitment) and α = 0.95 (ESCM-behavior commitment) [29]. In a confirmatory factor analysis, the model fit the data: χ^2^ = 258.35, *p* < 0.0001, GFI = 0.93, NFI = 0.95, TLI = 0.96, CFI = 0.97, SRMR = 0.036, and RMSEA (90% CI) = 0.08 [30].

This model consists of 13 items with two subscales: cognitive commitment (e.g., “My fists always get sweaty before a game”) and behavior commitment (e.g., “I constantly think about tactics while I am involved in a game”) [30]. The cognitive commitment subscale has seven questions, while the behavior commitment subscale has six questions, each arranged on a five-point Likert scale ranging from 1 (rarely) to 5 (almost always). Higher total scores represent a higher tendency of sports commitment.

#### 2.2.3. Golf Handicap

Golf handicap in both professional and amateur golfers was calculated in two steps based on the World Handicap System [31]. Step 1 was to calculate the lowest 10 of the player’s last 20 rounds of official KGA games using the formula: (score−course rating)×113 ÷slope rating. In Step 2, with the 10 lowest (i.e., best) scores, the handicap was calculated as: sum of 10 lowest scores÷10×0.96.

###  2.3. Analysis and Statistics

Statistical analysis was performed using SPSS for Windows (ver. 18.0; SPSS Inc., Chicago, IL, USA). All data were tested for normality using a Kolmogorov–Smirnov test (K-S test), and by calculating skewness and kurtosis. Skewness is a measure of asymmetry of the probability distribution [32] while kurtosis is a measure of the tailedness of the probability distribution [33]. Skewness and kurtosis values ranging between −2 and +2 are considered acceptable with respect to the data being normally distributed [34].

Age, golf career length (years), training time, and golf handicap between the two groups were analyzed with independent *t*-tests. The effect size of Cohen’s *d* for independent *t*-tests was interpreted as follows: 0.0 < *d* < 0.2, small; 0.3 < *d* < 0.5, medium; *d* > 0.6, large [35]. Sex distribution between the two groups was analyzed with a chi-squared test. The effect size of Cramer’s V for chi-squared tests was interpreted as follows: *V* > 0, no or very weak; *V* > 0.05, weak; *V* > 0.10, moderate; *V* > 0.15, medium; and *V* > 0.25, very strong [36].

Controlling for age, golf career length, and training time, the MPS and ESCM scores between the two groups were analyzed with ANCOVA tests. The effect size of partial eta-squared for ANCOVA was interpreted as follows: partial ŋ^2^ = 0.01–0.09, small; ŋ^2^ = 0.09–0.25, medium; and ŋ^2^ > 0.25, large [37].

To examine the relationship between golf career length and MPS in both professional and amateur golfers, partial correlations, controlling for age and training time, were used. To examine the relationship between golf handicap and MPS in both professional and amateur golfers, partial correlations, controlling for age, golf career length, and training time, were used.

Controlling for age, golf career length, and training time, partial correlations were used to examine the relationship between MPS and ESCM in both professional and amateur golfers. The correlation coefficient effect size (Pearson’s r) was interpreted as follows: 0.1 < *r* < 0.2, small; 0.3 < *r* < 0.5, moderate; and *r* > 0.6, large [35]. Statistical significance was set at *p* ≤ 0.05.

## 3. Results

### 3.1. Demographic and Psychological Characteristics

#### 3.1.1. Testing for Normality of Data

All data including age (professional golfer: K-S test statistic (D) = 0.08, *p* = 0.039, skewness *z* = 1.74, kurtosis *z* = 1.01; amateur golfer: D = 0.11, *p* = 0.45, skewness *z* = −1.10, kurtosis *z* = 1.39), golf career length (professional golfer: D = 0.15, *p* = 0.09, skewness *z* = 1.63, kurtosis *z* = 1.16; amateur golfer: D = 0.16, *p* = 0.08, skewness *z* = −1.03, kurtosis *z* = −1.01), training time (professional golfer: D = 0.12, *p* = 0.40, skewness *z* = 0.72, kurtosis *z* = 1.15; amateur golfer: D = 0.14, *p* = 0.55, skewness *z* = 0.78, kurtosis *z* = 0.65), golf handicap (professional golfer: D = 0.15, *p* = 0.52, skewness *z* = −0.67, kurtosis *z* = −0.53; amateur golfer: D = 0.12, *p* = 0.79, skewness *z* = 1.05, kurtosis *z* = 1.09), MPS total score (professional golfer: D = 0.18, *p* = 0.26, skewness *z* = -0.61, kurtosis *z* = −0.48; amateur golfer: D = 0.13, *p* = 0.69, skewness *z* = −1.15, kurtosis *z* = 1.02), commitment-Cognition score (professional golfer: D = 0.18, *p* = 0.29, skewness *z* = −1.24, kurtosis *z* = 1.01; amateur golfer: D = 0.13, *p* = 0.36, skewness *z* = 0.11, kurtosis *z* = −0.29), and commitment-Behavior score (professional golfer: D = 0.13, *p* = 0.30, skewness *z* = −0.33, kurtosis *z* = 0.36; amateur golfer: D = 0.10, *p* = 0.86, skewness *z* = 0.67, kurtosis *z* = −0.93) were slightly skewed and kurtotic for both professional and amateur golfers, but they did not differ significantly from a normal distribution.

#### 3.1.2. Comparison of Demographic Data between Professional Golfers and Amateur Golfers

There was no significant difference in the sex distribution between the two groups (χ^2^ = 0.8, *p* = 0.34, effect size [ES] = 0.035, small ES). Professional golfers were younger (−102.7%, *t* = 45.9, *p* < 0.01, ES = 3.012, large ES), and had fewer golf career years (−55.6%, *t* = 8.7, *p* < 0.01, ES = 0.592, medium ES) and lower golf handicap (−13.9%, *t* = 15.1, *p* < 0.01, ES = 1.308, large ES), compared to amateur golfers. Professional golfers showed longer training time, compared to amateur golfers (50.7%, *t* = 13.3, *p* < 0.01, ES = 1.293).

#### 3.1.3. Comparison of Perfectionism and Commitment between Professional Golfers and Amateur Golfers

Professional golfers had lower MPS total scores (−4.7%, *F* = 4.8, *p* = 0.02, ES = 0.341, large ES), as well as lower MPS self-oriented scores (−10.3%; *F* = 8.9, *p* < 0.01, ES = 0.498, large ES), compared to amateur golfers. In addition, professional golfers showed lower MPS other-oriented scores at a trend level compared to amateur golfers (−1.2%, *F* = 3.7, *p* = 0.06, ES = 0.224, moderate ES). Finally, professional golfers showed higher commitment-Cognition scores (12.6%, *F* = 9.4, *p* < 0.01, ES = 0.691, large ES) and commitment-Behavior scores (9.4%, *F* = 4.6, *p* = 0.03, ES = 0.479, large ES) compared to amateur golfers.

### 3.2. Correlation between Perfectionism, Commitment, Golf Handicap and Golf Career Length

#### 3.2.1. Correlation between Perfectionism and Commitment

Professional golfers’ MPS total scores were negatively associated with commitment-Cognition scores (*r* = −0.30, *p* < 0.01, −0.388 < 95% confidence interval [CI] < −0.228, moderate ES), while amateur golfers did not show a correlation between MPS scores and commitment-Cognition (*r* = 0.07, *p* = 0.61, −0.108 < 95% CI < 0.1485, small ES). In both professional (*r* = 0.15, *p* = 0.01, 0.032 < 95% CI < 0.206, small ES) and amateur golfers (*r* = −0.17, *p* < 0.01, −0.216 < 95% CI < −0.042, small ES), commitment-Cognition scores were not associated with golf handicap (Figure 1).

#### 3.2.2. Correlation between Perfectionism, Golf Handicap and Golf Career Length

In both professional golfers (*r* = −0.38, *p* < 0.01, −0.472 < 95% CI < −0.323, moderate ES) and amateur golfers (*r* = −0.53, *p* < 0.01, −0.642 < 95% CI < −0.465, large ES), golf career length was negatively correlated with golf handicap. In professional golfers, the golf handicap scores were positively correlated with MPS total scores (*r* = 0.33, *p* < 0.01, 0.281 < 95% CI < 0.435, moderate ES), MPS other-oriented scores (*r* = 0.34, *p* < 0.01, 0.281 < 95% CI < 0.435, moderate ES), and MPS socially prescribed scores (*r* = 0.37, *p* < 0.01, 0.312 < 95% CI < 0.462, moderate ES). Amateur golfers did not show a correlation between MPS total scores and the golf handicap scores (*r* = 0.12, *p* = 0.02, 0.002 < 95% CI < 0.253, small ES) (Figure 1).

## 4. Discussion

To the best of our knowledge, this study is the first of its kind to show a correlation between perfectionist traits and commitment in golfers. This study aimed to investigate differences in perfectionist traits and commitment between professional and amateur golfers using the MPS and ESCM. In particular, we focused on whether the level of perfectionist traits in professional golfers correlated with golf handicap and level of commitment. The current study showed that perfectionism and commitment were associated with golfers’ level of play (professional vs. amateur). Factors of perfectionism and commitment were also found to affect golf handicap. In addition, high-level performance in golfers was marked by lower perfectionism and high commitment during competition.

In the current study, professional golfers were younger, and, while they had fewer golf career years, they also had a lower golf handicap, compared to amateur golfers. In both professional and amateur golfers, golf career length was negatively correlated with golf handicap. Not surprisingly, this indicates that professional golfers may be more talented than amateur golfers. Professional golfers also showed longer training time than amateur golfers. Taken together, professional golfers exercised harder than amateur golfers although professional golfers were more talented than amateur golfers.

In a comparison of professional and amateur golf players, perfectionism scores (overall and as related to oneself and to others) were found to be higher in the amateur golfers. These differences supported the theory that non-professional athletes would be associated with higher levels of perfectionistic strivings in sport. These results are especially interesting in light of the findings of Kang et al. [5] who reported that perfectionistic athletes felt threatened, which resulted in an increase in anxiety and the perception that evaluative situations were opportunities for failure. High perfectionists were also found to incur more stress and greater depression than their less perfectionist peers [38].

We also found that commitment scores were higher in professional golfers than in amateur golfers. Interestingly, both commitment subscales (Cognition and Behavior) were significantly different between professional and amateur golfers. The results of this study are consistent with those of Kang et al. [5], who reported that a group of individuals in the top 10% of the KLPGA had higher commitment scores than those in a non-KLPGA group. Those findings suggest that high-level elite athlete performance is moderately positively associated with commitment intensity. Similarly, these findings suggest that elite golfers perceived commitment to be at least potentially controllable, as has been the case with previous research [39,40].

Pearson’s correlation results supported our hypothesis that golf handicap in professional golfers was positively correlated with total scores for perfectionism, other-oriented, and socially prescribed perfectionism. In general psychology literature, people who scored highly in perfectionism were more likely to report negative thoughts across their life span [41,42]. In addition, athletes who scored highly in other-oriented and socially prescribed perfectionism were positively correlated with maladaptive outcomes in sports [7]. Prior research on perfectionism has identified both positive and negative aspects to this personality trait [43,44]. In particular, negative perfectionists are overly self-critical, rarely feel competent in carrying out their responsibilities and duties, and consistently doubt the quality of their performance [45]. This is because high obsessive traits have been found to increase the likelihood of perfectionist traits involving anxiety and distress [46].

In professional golfers, perfectionism scores were negatively correlated with the commitment-Cognition scores. In the general psychology literature, perfectionist traits are closely associated with overcommitment, which has been shown to lead to burnout and decreased performance in athletes [47]. Generally, performance in sports and exercise is closely associated with working memory, including during training and learning [48,49]. In addition, faster processing speeds involved in working memory capacity are positively correlated with a high commitment [18]. Working memory is defined as the retention of information over a brief period of time, a function that is of central importance for a wide range of cognitive tasks [50]. Working memory capacity can predict performance in sports that have a substantial mental component such as golf [51]. In addition to working memory, peak performance in elite golfers is associated with a narrowed focus of attention, commitment in the present, feelings of confidence, and appropriate strategy [52,53]. Moreover, Silver et al. noted that impaired working memory capacity can damage goal-oriented behavior, cause cognitive disorganization, a failure to self-monitor, and reduce processing speed [51,54]. Such examples of working memory deficits occur subtly as neurocognitive dysfunction in athletes [55,56]. As a result, athletic performance appears to decrease [55,56].

Although simple skills can be learned through repetitive practices, learning complex skills, such as developing one’s senses and awareness, can be more effective through individual (independent) sports [57]. Commitment plays an important role in one’s successful performance in sports by increasing intellectual capacity, allowing for accurate decisions to be made and consistent prediction in different situations [58]. In this context, elite golfers are more likely to have high commitment when planning strategies and having detailed plans during competitions. This was reflected in the current study as professional golfers putting in more training time than amateur golfers.

It was hypothesized that perfectionist traits in professional golfers would negatively correlate with levels of commitment, but positively correlate with golf handicap. Based on the results of the current study, high-level performance in golfers is marked by lower perfectionism and high commitment during competition. The negative correlation between perfectionism and commitment may have been due to psychological factors including stress, anxiety, and mood. Although we did not assess the psychological status of golfers in the current study, we can hypothesize as to the correlation between them. The effect of the mental aspect of performance in golf was more pronounced in higher levels of competition [59]. In addition, different levels of psychological status, including mood and anxiety, were associated with different levels of performance skill in golf [60]. Increased commitment to controlling anxiety could lead to enhanced athletic performance [61]. Those findings would suggest psychological training and performance enhancement in golfers. Controlling obsessive perfectionism in golfers may encourage the desire to participate in the sport, which can lead to improved performance reflected by an improvement in their golf handicap.

## 5. Limitations

Although the results of the current study contribute to an understanding of the relationship between perfectionism and commitment in golfers, the study has some limitations. One limitation is the cross-sectional nature of the study, which means causality between study variables cannot be inferred. Longitudinal research would help to address this limitation by determining whether perfectionist traits are associated with elite golfer’s commitment during competition. Second, other factors that affect performance in golfers, including sports stress, physical injuries, and psychological states, were not considered in this study. Finally, perfectionism is thought to be affected by cultural differences. Therefore, readers should be cautious in generalizing the current results. Further studies should focus on assessing factors such as sports stress, psychological factors, physical injuries, and cultural differences in golfers.

## 6. Conclusions

The present study showed that professional golfers had lower levels of perfectionism and higher levels of commitment than amateur golfers. In professional golfers, perfectionist traits were negatively correlated with levels of commitment, but positively correlated with golf handicap. Findings suggest that golfers seeking to attain high levels of performance must consider the mental aspect of golfing and find ways to maximize levels of commitment and minimize perfectionism traits.

## Figures and Tables

**Figure 1 ijerph-17-05657-f001:**
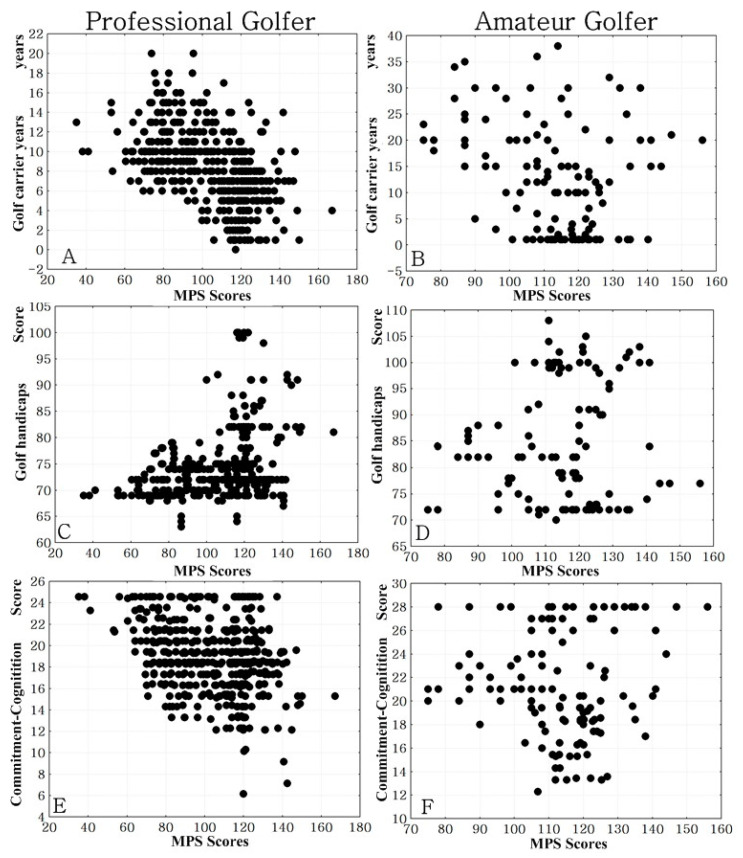
Scatter plot between perfectionism, commitment, golf handicap, and carrier years. MPS: Multidimensional Perfectionism Scale; ES: effect size: (**A**) controlling for age and training time, the partial correlations between golf career length (years) and MPS total scores in professional golfers, *r* = −0.38, *p* < 0.01, −0.472 < 95% CI < −0.323, moderate ES; (**B**) controlling for age and training time, the partial correlations between golf career length (years) and MPS total scores in amateur golfers, *r* = −0.53, *p* < 0.01, −0.642 < 95% CI < −0.465, large ES; (**C**) controlling for age, golf handicap, and training time, the partial correlations between golf handicap and MPS total scores in professional golfers, *r* = 0.33, *p* < 0.01, 0.281 < 95% CI < 0.435, moderate ES; (**D**) controlling for age and training time, the partial correlations between golf handicap and MPS total scores in amateur golfers, *r* = 0.12, *p* = 0.02, 0.002 < 95% CI < 0.253, small ES; (**E**) controlling for age, golf handicap, and training time, the partial correlations between commitment-Cognition scores and MPS total scores in professional golfers, *r* = −0.30, *p* < 0.01, −0.388 < 95% CI < −0.228, moderate ES; and (**F**) controlling for age, golf handicap, and training time, the partial correlations between commitment-Cognition scores and MPS total scores in amateur golfers, *r* = 0.07, *p* = 0.61, −0.108 < 95% CI < 0.1485, small ES.

**Table 1 ijerph-17-05657-t001:** Demographic and psychological characteristics of professional golfers vs. amateur golfers.

	Professional Golfers (*n* = 486)	Amateur Golfers (*n* = 233)	Percentage of Change	Statistics
Age (years) ^†^	22.1 ± 3.0	44.8 ± 10.2	102.7	*t* = 45.9, *p* < 0.01, ES = 3.012
Sex (male/female) ^§^	233/253	103/130		χ^2^ = 0.8, *p* = 0.34, ES = 0.035
Golf career (years) ^†^	8.1 ± 3.5	12.6 ± 10.2	55.6	*t* = 8.7, *p* < 0.01, ES = 0.592
Training time (hours/day) ^†^	6.5 ± 2.6	3.2 ± 2.5	50.7	*t* = 13.3, *p* < 0.01, ES = 1.293
Golf handicap ^†^	73.7 ± 5.8	84.0 ± 11.1	13.9	*t* = 15.1, *p* < 0.01, ES = 1.308
MPS total ^††^	107.7 ± 20.9	112.8 ± 15.6	4.7	*F =* 4.8, *p* = 0.02, ES = 0.560
MPS Self-oriented ^††^	38.9 ± 6.5	42.0 ± 9.0	10.3	*F =* 8.9, *p* < 0.01, ES = 0.598
MPS Others-oriented ^††^	33.3 ± 7.5	34.4 ± 7.9	3.2	*F =* 3.7, *p* = 0.06, ES = 0.587
MPS Socially prescribed ^††^	34.4 ± 10.2	34.8 ± 7.9	1.2	*F =* 0.7, *p* = 0.41, ES = 0.069
Commitment-Cognition ^††^	21.4 ± 4.5	18.9 ± 3.4	12.6	*F =* 9.4, *p* < 0.01, ES = 0.691
Commitment-Behavior ^††^	17.4 ± 3.8	15.8 ± 2.8	9.4	*F =* 4.6, *p* = 0.03, ES = 0.479

MPS, Multidimensional Perfectionism Scale; ES, effect size; ^†^ Independent *t*-test; ^††^ ANCOVA with age, golf career length, and training time as covariates; ^§^ Chi-squared test.

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
