# Peer review of "The Comparison of Perfectionism and Commitment between Professional and Amateur Golfers and the Association between Perfectionism and Commitment in the Two Groups"

_ijerph, 2020, doi:10.3390/ijerph17165657_

Round 1

Reviewer 1 Report

The author aimed to investigate differences in perfectionism traits and commitment 9 between professional and amateur golfers who were members of the Korean Golf Association 10 (KGA).

in general, that paper is good but :

1- Abstract: Need to be more clearly 

2- MPS: Multidimensional Perfectionism Scale Need Figure more clearly

Author Response

Reviewer 1

RESPONSE: We revised misspellings in accordance with your comments.

Abstract: Need to be more clearly

RESPONSE: We added the values of the statistical results in the abstract.

MPS (Multidimensional Perfectionism Scale) need figure more clearly.

RESPONSE: As the reviewer suggested, we modified the MPS figure in the revised manuscript.

Reviewer 2 Report

Despite an interesting topic, there a major revision to conduct. Methods, results, and discussion must be deeply improved. Please consider some suggestions and comments.

Abstract

Lines 15-19: add values during the description of results. Examples “showed lowers scores” – how much? Add percentage of change and the p-value and effect size, changing to “showed lower scores than something (-x.xx%; p = x.xxx; ES = x.xxx)”

Introduction

Lines 25-26: support the statement with a reference

Lines 30-31: contextualize the study and also describing the population

Lines 49-42: maybe this paragraph can be part of the previous one, making the two in one single paragraph.

Lines 43-44: this sentence is a fragment. Please work on the link between paragraphs.

Lines 50-53: maybe the related work must be improved adding more studies and results that can improve the rationale and the justification for the current study.

Lines 54-56: Before the objective of the study, there is a need to add the statement of contribution section.

Overall: the introduction must be improved by adding stronger related work and a rationale that conducts from key-concepts to the main objective. I the middle, strongly related work, and statement of the contribution must appear to support the objective of the study.

Methods

Split the terms design and subjects. In fact, no information about the design was provided in the section. Thus, only information about participants was included.

Lines 65-68: detail the inclusion criteria

I recommend adding a sub-section of “an experimental approach to the problem”. In this sub-section, a first sentence will provide information about the study design. After that, a brief overview of the methodology will be detailed, aiming to increase the understanding of the approach to answer the objective of the study. Information about the procedures of application the questionnaires must be added.

Lines 77-80: provide information about the validity and reliability of the questionnaire. Moreover, items of the questionnaire can be added as a supplement to this article. This will help readers to understand the questionnaire.

Lines 85-87: provide information about the validity and reliability of the questionnaire.

Lines 94-95: parametric tests are being used without information of assumptions. Add values of normality and homogeneity obtained for the sample. Moreover, for the cases of analysis of variance, the effect size must be added. Some articles to support: Steiger, J. H. (2004). Beyond the F test: effect size confidence intervals and tests of close fit in the analysis of variance and contrast analysis. Psychological methods9(2), 164.; Wampold, B. E., & Serlin, R. C. (2000). The consequence of ignoring a nested factor on measures of effect size in analysis of variance. Psychological methods5(4), 425. And Ferguson, C. J. (2016). An effect size primer: a guide for clinicians and researchers.

Lines 97-98: interpretation of correlation magnitude and also effect size must be added. The thresholds must be also reported. You can check here: https://www.sportsci.org/resource/stats/effectmag.html

Results

In all the sections the effect size must be added after the p-value. Moreover, interpretation of the magnitude of changes must be also added (example: ES = x.xxx; moderate ES)

Table 1: add the units in the first column (example: age [years]). I also would like to recommend a new column with the percentage of change between factors (professional vs. amateur).

Report of correlations would benefit from adding intervals of confidence for the correlation coefficient.

Figure 1: add the units in the y- and x-axis. Maybe in the title may replace “correlation” by “scatter plot” that is the right information in this case.

Discussion

The discussion is not deep enough and informative enough. Recommendations: (i) after remembering the objective of the study, add a synthesizes of the findings; (ii) per each category of analysis produce a consistent paragraph presenting the evidence, comparing to previous studies and justifying the results considering the theoretical background; (iii) present the clinical relevance of the findings; (iv) present recommendation for future studies after the study limitations; (v) add a section of practical implications.

Author Response

Reviewer 2

Abstract

Lines 15-19: add values during the description of results. Examples “showed lowers scores” – how much? Add percentage of change and the p-value and effect size, changing to “showed lower scores than something (-x.xx%; p = x.xxx; ES = x.xxx)”

RESPONSE: We added the statistical values as follows:

“An ANCOVA, controlling for age, career length, and training time, revealed lower MPS self-oriented scores (-10.3%; F = 8.9, p < 0.01; Å‹2 = 0.498), and higher ESCM-Cognition (12.6%; F = 9.4, p < 0.01; Å‹2 = 0.691) and ESCM-Behavior (9.4%; F = 4.6, p = 0.03; Å‹2 = 0.479) scores in professional golfers than amateur golfers. In partial correlations controlling for age, career length, and training time, MPS scores were negatively associated with ESCM-Cognition scores (r = -0.31, p < 0.001) only in professional golfers. Professional golfers’ mean handicaps were positively correlated with MPS total (r = 0.36, p < 0.01).”

Introduction (Lines 25-26): Support the statement with a reference.

RESPONSE: We added the reference to support this statement as follows:

“Perfectionism is a personality trait characterized by striving toward flawlessness and setting excessively high standards for performance (Stoeber et al., 2007).”

Introduction (Lines 30-31): Contextualize the study and also describing the population.

RESPONSE: We added text describing the study and population as follows:

“In a study of 11 female patients and 11 matched healthy controls, van der Wee et al. [7] reported that obsessive traits would lead to increased deficits in response to higher levels of task difficulty in a working memory task.”

Introduction (Lines 39-42): maybe this paragraph can be part of the previous one, making the two in one single paragraph.

RESPONSE: As the reviewer suggested, we made the two paragraphs into one single paragraph.

Introduction (Lines 43-44): this sentence is a fragment. Please work on the link between paragraphs.

RESPONSE: We revised the sentences as follows:

“Conversely, improved working memory capacity seems to correlate with an athletes’ highest level of skill [17–23]. Numerous studies have suggested that improved working memory is highly associated with processing speed [17–20].”

Introduction (Lines 50-53): maybe the related work must be improved adding more studies and results that can improve the rationale and the justification for the current study.

RESPONSE: We added additional studies related to commitment. The section was modified as follows:

“Sport commitment represents a psychological state reflecting an athlete’s desire and resolve to continue his or her sport participation [25], and there is empirical support for the view that high levels of sport commitment accompany greater behavioral persistence [26]. Numerous studies have suggested that commitment is associated with high level performance or peak experience in sports [27]. Swann et al. [28] reported that commitment in elite golfers including pre-shot routines, psychological interventions, and a healthy physical state were important factors for good performance.”

Introduction (Lines 54-56): Before the objective of the study, there is a need to add the statement of contribution section.

RESPONSE: As the reviewer suggested, we added a statement of contribution section as follows:

“As mentioned above, results elucidating the relationship between perfectionism and commitment as related to an athlete’s performance could aid golfers who struggle with perfectionism. With such research in mind, this study aimed to investigate the differences in perfectionist traits and commitment between professional and amateur golfers who were members of the Korean Golf Association (KGA).”

Methods (Lines 65-68): Detail the inclusion criteria.

RESPONSE: As the reviewer suggested, we added details regarding the inclusion criteria as well as information about the study design. The revised section is as follows:

“This study was designed as a cross-sectional case control study with two groups. All participants were recruited using simple random sampling. Participants included 486 professional golf athletes (mean age± standard deviation = 22.1 ± 3.0 years) and 233 amateur golf athletes (44.8 ± 10.2 years) who were members of the KGA.

The number of male and female professional golfers was 233 (47.9%) and 253 (52.1%), respectively, and amateur golfers was 103 (44.2%) and 130 (55.8%), respectively. The average career length of professional golfers and amateur golfers was 8.1 ± 3.5 and 12.6 ± 10.2 years, respectively. The average training time of professional and amateur golfers was 6.5 ± 2.6 and 3.2 ± 2.5 hours per day, respectively. The average scores of golf games (handicaps) of professional golfers and amateur golfers was 73.7 ± 5.8 and 84.0 ± 11.1, respectively.

Participants in the professional group met the following inclusion criteria: (i) Division 1 golf players in the KGA; (ii) aged from 18 to 60 years, and (iii) more than 2 years of experience in the Korea Professional Golfers Association (KPGA) or Korea Ladies Professional Golfers Association (KLPGA). Participants in the amateur group met the following inclusion criteria: (i) participated in at least one amateur tournament per year in Korea in the previous 2 years, (ii) aged from 18 to 60 years, and (iii) no history of KPGA or KLPGA membership (Table 1).

All athletes were assessed with the Multidimensional Perfectionism Scale (MPS) and the Expansion of Sports Commitment Model (ESCM). Data from 18 professional athletes and 11 amateur athletes were excluded due to incomplete answers or large amounts of missing data. Participants gave written, informed consent to participate in the study. All data were collected on the golf courses of the above-mentioned tournament sites, with full cooperation of the KGA. Specifically, after the players arrived at the club house, the survey was conducted before they entered the locker room. The protocol of this study was approved by the Chung Ang University Hospital Review Board (C2014111).”

Methods (Lines 65-68): Provide information about the validity and reliability of questionnaire. Moreover, items of the questionnaire can be added as a supplement to this article. This will help readers to understand the questionnaire.

RESPONSE: We revised the text as follows:

“This 45-question scale has three dimensions of self-oriented (e.g., “When I am working on something, I cannot relax until it is perfect”), other-oriented (e.g., “I am not likely to criticize someone for giving up too easily”), and socially prescribed perfectionism (e.g., “I find it difficult to meet others’ expectations of me”). A Korean version of the MPS has been verified in Korean university students [31] with a reliability of α = 0.84 (self-oriented), α = 0.73 (other-oriented), and α = 0.76 (socially prescribed). Each dimension has 15 questions arranged on a 7-point Likert scale ranging from 1 (completely disagree) to 7 (completely agree).”

Methods (Lines 85-87): Provide information about the validity and reliability of the questionnaire.

RESPONSE: We revised the text as follows:

This model consists of 13 items with two subscales: cognitive commitment (e.g., “My fists always get sweaty before a game”) and behavior commitment (e.g., “I constantly think about tactics while I am involved in a game”) [32]. The cognitive commitment subscale has seven questions, while the behavior commitment subscale has six questions, each arranged on a 5-point Likert scale ranging from 1 (rarely) to 5 (almost always). The ESCM-Korean version has been verified in Korean sports participants with reliability scores of α = 0.86 (ESCM-cognitive commitment) and α = 0.95 (ESCM-behavior commitment).”

Methods (Lines 94-95): Parametric tests are being used without information of assumptions. Add values of normality and homogeneity obtained for the sample. Moreover, for the cases of analysis of variance, the effect size must be added.

RESPONSE: We added details regarding testing for normality in the Methods section (section 2.3. Analysis and Statistics) and the values were added to the Results section.

“Using a Kolmogorov-Smirnov test, all data were tested for normality.”

“All data including age (professional golfer: skewness z = 1.74, kurtosis z = 1.01; amateur golfer: skewness z = -1.10, kurtosis z = 1.39), career length (professional golfer: skewness z = 1.63, kurtosis z = 1.16; amateur golfer: skewness z = -1.03, kurtosis z = -1.01), training time (professional golfer: skewness z = 0.72, kurtosis z = 1.15; amateur golfer: skewness z = 0.78, kurtosis z = 0.65), handicap (professional golfer: skewness z = -0.67, kurtosis z = -0.53; amateur golfer: skewness z = 1.05, kurtosis z = 1.09), MPS total score (professional golfer: skewness z = -0.61, kurtosis z = -0.48; amateur golfer: skewness z =-1.15, kurtosis z = 1.02), commitment-Cognition score (professional golfer: skewness z = -1.24, kurtosis z = 1.01; amateur golfer: skewness z = 0.11, kurtosis z = -0.29), and commitment-Behavior score (professional golfer: skewness z = -0.33, kurtosis z = 0.36; amateur golfer: skewness z = 0.67, kurtosis z = -0.93) were slightly skewed and kurtotic for both professional and amateur golfers, but they did not differ significantly from normality.”

→ For homogeneity of the sex distribution between the two groups, we included results from a chi-square test in the Results.

“There was no significant difference in the sex distribution between the two groups (χ2 = 0.8, p = 0.34, effect size [ES] = 0.035, small ES).”

Methods (Lines 97-98): Interpretation of correlation magnitude and also effect size must be added. The thresholds must be also reported.

RESPONSE: We added the interpretation of the magnitude of the correlation and effect size in the Methods section.

 “The correlation coefficient effect size (Pearson’s r) was interpreted as follows: 0.1 < r < 0.2, small; 0.3 < r < 0.5, moderate; and r > 0.6, large.”

Results_ In all the sections the effect size must be added after the p-value. Moreover, interpretation of the magnitude of changes must be also added (example: ES = x.xxx; moderate ES).

RESPONSE: We added effect sizes and the magnitude of change to all of the results.

There was no significant difference in the sex distribution between the two groups (χ2 = 0.8, p = 0.34, effect size [ES] = 0.035, small ES). Professional golfers were younger (-102.7%, t = 45.9, p < 0.01, ES = 3.012, large ES), and had fewer career years (-55.6%, t = 8.7, p < 0.01, ES = 0.592, medium ES) and lower handicaps (-13.9%, t = 15.1, p < 0.01, ES = 1.308, large ES), compared to amateur golfers. Professional golfers showed longer training time, compared to amateur golfers (50.7%, t = 13.3, p < 0.01, ES = 1.293).

Professional golfers had lower MPS total scores (-4.7%, F = 4.8, p = 0.02, ES = 0.341, large ES), as well as lower MPS self-oriented scores (-10.3%; F = 8.9, p < 0.01, ES = 0.498, large ES), compared to amateur golfers. Also, professional golfers showed lower MPS other-oriented scores at a trend level compared to amateur golfers (-1.2%, F = 3.7, p = 0.06, ES = 0.224, moderate ES). Finally, professional golfers showed higher commitment-Cognition scores (12.6%, F = 9.4, p < 0.01, ES = 0.691, large ES) and commitment-Behavior scores (9.4%, F = 4.6, p = 0.03, ES = 0.479, large ES) compared to amateur golfers.

Correlation Between Perfectionism, Commitment, Handicap, and Career Length

Professional golfers’ MPS total scores were negatively associated with commitment-Cognition scores (r = -0.31, p < 0.01, -0.388 < 95% confidence interval [CI] < -0.228, moderate ES), while amateur golfers did not show a correlation between MPS scores and commitment-Cognition (r = 0.02, p = 0.82, -0.108 < 95% CI < 0.1485, small ES). In both professional (r = 0.12, p = 0.17, 0.032 < 95% CI < 0.206, small ES) and amateur golfers (r = -0.13, p < 0.01, -0.216 < 95% CI < -0.042, small ES), commitment-Cognition scores were not associated with handicap.

In both professional golfers (r = -0.40, p < 0.01, -0.472 < 95% CI < -0.323, moderate ES) and amateur golfers (r = -0.56, p < 0.01, -0.642 < 95% CI < -0.465, large ES), career length was negatively correlated with handicap. In professional golfers, handicaps were positively correlated with MPS total scores (r = 0.36, p < 0.01, 0.281 < 95% CI < 0.435, moderate ES), MPS other-oriented scores (r = 0.36, p < 0.01, 0.281 < 95% CI < 0.435, moderate ES), and MPS socially prescribed scores (r = 0.39, p < 0.01, 0.312 < 95% CI < 0.462, moderate ES). Amateur golfers did not show a correlation between MPS total scores and handicaps (r = 0.13, p = 0.15, 0.002 < 95% CI < 0.253, small ES).”

Results_ Table 1: Add the units in the first column (example: age [years]). I also would like to recommend a new column with the percentage of change between factors (professional vs. amateur).

RESPONSE: We revised Table 1 as follows:

Table 1. Demographic and psychological characteristics of professional golfers vs. amateur golfers.

Professional golfers

(n = 486)

Amateur golfers

(n = 233)

Percentage of change

Statistics

Age [years]

22.1 ± 3.0

44.8 ± 10.2

102.7

t = 45.9, p < 0.01, ES = 3.012

Sex [male/female] §

233/253

103/130

χ2 = 0.8, p = 0.34, ES = 0.035

Career [years]

8.1 ± 3.5

12.6 ± 10.2

55.6

t = 8.7, p < 0.01, ES = 0.592

Training time [hours/day]

6.5 ± 2.6

3.2 ± 2.5

50.7

t = 13.3, p < 0.01, ES = 1.293

Handicap

73.7 ± 5.8

84.0 ± 11.1

13.9

t = 15.1, p < 0.01, ES = 1.308

MPS total ††

107.7 ± 20.9

112.8 ± 15.6

4.7

F = 4.8, p = 0.02, ES = 0.560

MPS Self-oriented ††

38.9 ± 6.5

42.0 ± 9.0

10.3

F = 8.9, p < 0.01, ES = 0.598

MPS Others-oriented ††

33.3 ± 7.5

34.4 ± 7.9

3.2

F = 3.7, p = 0.06, ES = 0.587

MPS Socially prescribed ††

34.4 ± 10.2

34.8 ± 7.9

1.2

F = 0.7, p = 0.41, ES = 0.069

Commitment-Cognition ††

21.4 ± 4.5

18.9 ± 3.4

12.6

F = 9.4, p < 0.01, ES = 0.691

Commitment-Behavior ††

17.4 ± 3.8

15.8 ± 2.8

9.4

F = 4.6, p = 0.03, ES = 0.479

MPS: Multidimensional Perfectionism Scale; Independent t-test; †† ANCOVA with age, career length, and training time as covariates; § Chi-squared test, effect size (ES)

Results_Report of correlations would benefit from adding intervals of confidence for the correlation coefficient.

RESPONSE: We added confidence intervals for the correlation coefficients in the Results section.

Professional golfers’ MPS total scores were negatively associated with commitment-Cognition scores (r = -0.31, p < 0.01, -0.388 < 95% confidence interval [CI] < -0.228, moderate ES), while amateur golfers did not show a correlation between MPS scores and commitment-Cognition (r = 0.02, p = 0.82, -0.108 < 95% CI < 0.1485, small ES). In both professional (r = 0.12, p = 0.17, 0.032 < 95% CI < 0.206, small ES) and amateur golfers (r = -0.13, p < 0.01, -0.216 < 95% CI < -0.042, small ES), commitment-Cognition scores were not associated with handicap.

In both professional golfers (r = -0.40, p < 0.01, -0.472 < 95% CI < -0.323, moderate ES) and amateur golfers (r = -0.56, p < 0.01, -0.642 < 95% CI < -0.465, large ES), career length was negatively correlated with handicap. In professional golfers, handicaps were positively correlated with MPS total scores (r = 0.36, p < 0.01, 0.281 < 95% CI < 0.435, moderate ES), MPS other-oriented scores (r = 0.36, p < 0.01, 0.281 < 95% CI < 0.435, moderate ES), and MPS socially prescribed scores (r = 0.39, p < 0.01, 0.312 < 95% CI < 0.462, moderate ES). Amateur golfers did not show a correlation between MPS total scores and handicaps (r = 0.13, p = 0.15, 0.002 < 95% CI < 0.253, small ES).”

Results_Figure 1: add the units in the y- and x-axis. Maybe in the title may replace “correlation” by “scatter plot” that is the right information in this case.

RESPONSE: We revised Figure 1 and the figure legend as follows (in PDF file):

MPS: Multidimensional Perfectionism Scale; ES: effect size. (a) Controlling for age and training time, the partial correlations between career length (years) and MPS total scores in professional golfers, r = -0.40, p < 0.01, -0.472 < 95% CI < -0.323, moderate ES; (b) Controlling for age and training time, the partial correlations between career length (years) and MPS total scores in amateur golfers, r = -0.56, p < 0.01, -0.642 < 95% CI < -0.465, large ES; (c) Controlling for age, handicap, and training time, the partial correlations between handicaps and MPS total scores in professional golfers, r = 0.36, p < 0.01, 0.281 < 95% CI < 0.435, moderate ES; (d) Controlling for age and training time, the partial correlations between handicaps and MPS total scores in amateur golfers, r = 0.13, p = 0.15, 0.002 < 95% CI < 0.253, small ES; (e) Controlling for age, handicap, and training time, the partial correlations between commitment-Cognition scores and MPS total scores in professional golfers, r = -0.31, p < 0.01, -0.388 < 95% CI < -0.228, moderate ES; (f) Controlling for age, handicap, and training time, the partial correlations between commitment-Cognition scores and MPS total scores in amateur golfers, r = 0.02, p = 0.82, -0.108 < 95% CI < 0.1485, small ES.

Discussion

The discussion is not deep enough and informative enough. Recommendations: (i) after remembering the objective of the study, add a synthesizes of the findings; (ii) per each category of analysis produce a consistent paragraph presenting the evidence, comparing to previous studies and justifying the results considering the theoretical background; (iii) present the clinical relevance of the findings; (iv) present recommendation for future studies after the study limitations; (v) add a section of practical implications.

RESPONSE: We revised the text in the Discussion section based on your recommendations as follows:

  • after remembering the objective of the study, add a synthesizes of the findings

RESPONSE: As you recommended, we revised the first paragraph as follows:

" To the best of our knowledge, this study is the first of its kind to show a correlation between perfectionist traits and commitment in golfers. This study aimed to investigate differences in perfectionist traits and commitment between professional and amateur golfers using the MPS and ESCM. In particular, we focused on whether the level of perfectionist traits in professional golfers correlated with handicap and level of commitment. The current study showed that perfectionism and commitment were associated with golfers' level of play (professional vs. amateur). Factors of perfectionism and commitment were also found to affect golfers’ handicap. In addition, high-level performance in golfers was marked by lower perfectionism and high commitment during competition.”

  • per each category of analysis produce a consistent paragraph presenting the evidence, comparing to previous studies and justifying the results considering the theoretical background

RESPONSE: For the explanation of each category of analysis, we added text as follows:

“In a comparison of professional and amateur golf players, perfectionism scores (overall and as related to oneself and to others) were found to be higher in the amateur golfers. These differences supported the theory that non-professional athletes would be associated with higher levels of perfectionistic strivings in sport. These results are especially interesting in light of the findings of Kang et al. [33] who reported that perfectionistic athletes felt threatened, which resulted in an increase in anxiety and the perception that evaluative situations were opportunities for failure. High perfectionists were also found to incur more stress and experience greater depression than their less perfectionist peers [34].

We also found that commitment scores were higher in professional golfers than in amateur golfers. Interestingly, both commitment subscales (Cognition and Behavior) were significantly different between professional and amateur golfers. The results of this study are consistent with Kang et al. [33] who reported that a group of individuals in the top 10% of the KLPGA had higher commitment scores than those in a non-KLPGA group. These findings suggest that high level elite athlete performance is moderately positively associated with commitment intensity. Similarly, these findings suggest that elite golfers perceived commitment to be at least potentially controllable, as has been the case with previous research [35,36].”

  • present the clinical relevance of the findings

RESPONSE: We added clinical implications of the findings in the Discussion.

“These findings would suggest psychological training and performance enhancement in golfers. Controlling obsessive perfectionism in golfers may encourage the desire to participate in the sport, which can lead to improved performance reflected by an improvement in their handicap.”

  • present recommendation for future studies after the study limitations

RESPONSE: This too was added to the Discussion.

“Further studies should focus on assessing factors such as sports stress, psychological factors, physical injuries, and cultural differences in golfers.”

Reviewer 3 Report

The paper has great potential and research of golfers are rarely found. However, the paper has some serious mistakes.

Introduction

The authors write "With such research in mind, this study aimed to investigate the differences in perfectionist traits and commitment between professional and amateur golfers who were members of the Korean Golf Association (KGA). In particular, we investigated whether the level of perfectionist traits in professional golfers correlated with their levels of commitment and handicaps (i.e., average score). "

The authors did not explain enough why they decided to choose two such different content dependent variables as commitment and handicap?

Due to the purpose of the study, authors should be better prepared for the relations presented later. In particular, it is necessary to justify why it is worth examining the relationship between perfectionism and commitment. It is also necessary to present other research results in which the relationship of perfectionism with commitment and level of sport (e.g. in other sports) was examined.

Methods

Subject

A more detailed description of the groups is needed, e.g. the sex of the respondents, sports level, years of training.

Measurments

Authors should define handicap and its indicators.

Analysis and Statistics

The authors write, among others, that they used ANOVA and post-hoc tests. In addition, the description of the analysis shows that four groups were analyzed, while the Results show a comparison of two groups only (professional and amateur golfers).

In standard way, to compare two groups, not ANOVA but the t test for independent data is used , although the p value will be the same as for F test in such cases.

No post hoc tests are performed when comparing two groups. Post - hoc tests are done for a significant F value when comparing at least three groups with each other.

Results

The authors write "Professional golfers showed lower MPS total scores compared to amateur golfers (F = 4.8, p = 0.02). In post hoc testing, professional golfers showed lower MPS self-oriented scores compared to amateur golfers (F = 8.9, p <0.01). "   The value of F from table 1 for „Self-oriented” is given in brackets. It is difficult to understand.

No effect size was given for presented comparisons.

In addition, the small description under Table 1 shows that the authors have introduced covariates: age, carrier years, training time. Nowhere else has it been written that ANCOVA was performed.

In addition, the although model is designated as ANOVA under the table covariates are provided (?).

There was a difference in  the groups  for additional variables (eg age)  and doing ANCOVA would be justified.

In addition, data on gender, career years, training time and hendicap are provided in this chapter. These data should be included in the characteristics of subject.

In the correlation analyzes, the relationship between perfectionism and career years was presented first, while the main goal was asses of relationship between perfectionism and commitment as well as perfectionism and handicap. The reader has not been ealier prepared for presented the relationship between perfectionism and carrier years.

The authors write "In both professional (r = -0.40, p <0.001) and amateur golfers (r = 0.12, p = 0.17), commitment-Cognition scores were not associated with handicap (r = -0.13, p <0.001)" The content of the sentence is not adequate to the given values of p.

The authors present correlations for two groups. To compare them, you need to test for the significance of differences for correlation coefficients in two different groups.

Discussion

In the discussion, the authors should refer more to the results. The authors should try to interpret the differences / similarities obtained in comparisons two groups. Eg why the correlations between MPS and Commitment differ between professionals and amateurs.

Author Response

Reviewer 3

Introduction

The authors did not explain enough why they decided to choose two such different content dependent variables as commitment and handicap? Due to the purpose of the study, authors should be better prepared for the relations presented later. In particular, it is necessary to justify why it is worth examining the relationship between perfectionism and commitment. It is also necessary to present other research results in which the relationship of perfectionism with commitment and level of sport (e.g. in other sports) was examined.

RESPONSE: In this revision, we further explained why we chose commitment and handicap as our dependent variables as follows:

As mentioned above, results elucidating the relationship between perfectionism and commitment as related to an athlete’s performance could aid golfers who struggle with perfectionism. With such research in mind, this study aimed to investigate the differences in perfectionist traits and commitment between professional and amateur golfers who were members of the Korean Golf Association (KGA). In addition, we investigated whether the level of perfectionist traits in professional golfers was related to performance, usually represented by average golf scores (i.e., handicaps). Athletes and coaches readily acknowledge the importance of the mental aspects of the game of golf, particularly at the highest levels of competition [29]. Psychological differences have been found between golfers of different skill levels, indicating that not only is the mental aspect of golf important, it may play a role in separating one level of play from another [30].

We hypothesized that perfectionist traits would negatively correlate with levels of commitment, but would positively correlate with golf performance, as measured by average scores, handicaps. In summary, we expected 'low perfectionist traits' to be associated with excellent performance (i.e., lower handicap) during competition.”

Method_ Subject: A more detailed description of the groups is needed, e.g. the sex of the respondents, sports level, years of training.

RESPONSE: We described the two groups in greater detail in section 2.1. Design and Subjects of the Methods as follows:

“The number of male and female professional golfers was 233 (47.9%) and 253 (52.1%), respectively, and amateur golfers was 103 (44.2%) and 130 (55.8%), respectively. The average career length of professional golfers and amateur golfers was 8.1 ± 3.5 and 12.6 ± 10.2 years, respectively. The average training time of professional and amateur golfers was 6.5 ± 2.6 and 3.2 ± 2.5 hours per day, respectively. The average scores of golf games (handicaps) of professional golfers and amateur golfers was 73.7 ± 5.8 and 84.0 ± 11.1, respectively.”

Method_ Measurements: Authors should define handicap and its indicators

RESPONSE: We added the definition of handicap as follows:

“Golfers’ Handicap

The handicap of professional golfers was defined as the average of their scores over the past year, which was calculated based on official records of official KGA games. The handicap of amateur golfers was defined as the average of their scores during golf games over the past year.”

The authors write, among others, that they used ANOVA and post-hoc tests. In addition, the description of the analysis shows that four groups were analyzed, while the Results show a comparison of two groups only (professional and amateur golfers).

In standard way, to compare two groups, not ANOVA but the t test for independent data is used , although the p value will be the same as for F test in such cases.

No post hoc tests are performed when comparing two groups. Post - hoc tests are done for a significant F value when comparing at least three groups with each other.

RESPONSE: We revised the statistical methods as follows:

“Analysis and Statistics

     Statistical analysis was performed using commercial software (SPSS for Windows, ver. 18.0; SPSS Inc., Chicago, IL, USA). Using a Kolmogorov-Smirnov test, all data were tested for normality. Age, career length (years), training time, and handicap between the two groups were analyzed with independent t-tests. The effect size of Cohen’s d for independent t-tests was interpreted as follows: 0.0 < d < 0.2, small; 0.3 < d < 0.5, medium; and d > 0.6, large. Sex distribution between the two groups was analyzed with a Chi-squared test. The effect size of Cramer’s V for Chi-squared tests was interpreted as follows: V > 0, no or very weak; V > 0.05, weak; V > 0.10, moderate; V > 0.15, medium; and V > 0.25, very strong.

Controlling age, career length, and training time, the MPS and ESCM scores between the two groups were analyzed with ANCOVA tests. The effect size of partial eta-squared for ANCOVA was interpreted as follows: partial Å‹2 = 0.01–0.09, small; Å‹2 = 0.09–0.25, medium; and Å‹2 > 0.25, large.

To examine the relationship between career length and MPS in both professional and amateur golfers, partial correlations, controlling for age and training time, were used. To examine the relationship between handicaps and MPS in both professional and amateur golfers, partial correlations, controlling for age, career length, and training time, were used.

Controlling for age, career length, and training time, partial correlations were used to examine the relationship between MPS and ESCM in both professional and amateur golfers. The correlation coefficient effect size (Cohen) was interpreted as follows: 0.1 < r < 0.2, small; 0.3 < r < 0.5, moderate; and r > 0.6, large. Statistical significance was set at p ≤ 0.05.”

Results_ "Professional golfers showed lower MPS total scores compared to amateur golfers (F = 4.8, p = 0.02). In post hoc testing, professional golfers showed lower MPS self-oriented scores compared to amateur golfers (F = 8.9, p <0.01). " The value of F from table 1 for “Self-oriented” is given in brackets. It is difficult to understand

RESPONSE: We revised Table 1 as follows:

Table 1. Demographic and psychological characteristics of professional golfers vs. amateur golfers.

Professional golfers

(n = 486)

Amateur golfers

(n = 233)

Percentage of change

Statistics

Age [years]

22.1 ± 3.0

44.8 ± 10.2

102.7

t = 45.9, p < 0.01, ES = 3.012

Sex [male/female] §

233/253

103/130

χ2 = 0.8, p = 0.34, ES = 0.035

Career [years]

8.1 ± 3.5

12.6 ± 10.2

55.6

t = 8.7, p < 0.01, ES = 0.592

Training time [hours/day]

6.5 ± 2.6

3.2 ± 2.5

50.7

t = 13.3, p < 0.01, ES = 1.293

Handicap

73.7 ± 5.8

84.0 ± 11.1

13.9

t = 15.1, p < 0.01, ES = 1.308

MPS total ††

107.7 ± 20.9

112.8 ± 15.6

4.7

F = 4.8, p = 0.02, ES = 0.560

MPS Self-oriented ††

38.9 ± 6.5

42.0 ± 9.0

10.3

F = 8.9, p < 0.01, ES = 0.598

MPS Others-oriented ††

33.3 ± 7.5

34.4 ± 7.9

3.2

F = 3.7, p = 0.06, ES = 0.587

MPS Socially prescribed ††

34.4 ± 10.2

34.8 ± 7.9

1.2

F = 0.7, p = 0.41, ES = 0.069

Commitment-Cognition ††

21.4 ± 4.5

18.9 ± 3.4

12.6

F = 9.4, p < 0.01, ES = 0.691

Commitment-Behavior ††

17.4 ± 3.8

15.8 ± 2.8

9.4

F = 4.6, p = 0.03, ES = 0.479

MPS: Multidimensional Perfectionism Scale; Independent t-test; †† ANCOVA with age, career length, and training time as covariates; § Chi-squared test, effect size (ES)

Results_ No effects size was given for presented comparisons.

RESPONSE: We added effect sizes in the Results section and in Table 1.

In addition, the small description under Table 1 shows that the authors have introduced covariates: age, carrier years, training time. Nowhere else has it been written that ANCOVA was performed.

RESPONSE: We added a description in section 2.1. Design and Subjects of the Methods as follows:

“The number of male and female professional golfers was 233 (47.9%) and 253 (52.1%), respectively, and amateur golfers was 103 (44.2%) and 130 (55.8%), respectively. The average career length of professional golfers and amateur golfers was 8.1 ± 3.5 and 12.6 ± 10.2 years, respectively. The average training time of professional and amateur golfers was 6.5 ± 2.6 and 3.2 ± 2.5 hours per day, respectively. The average scores of golf games (handicaps) of professional golfers and amateur golfers was 73.7 ± 5.8 and 84.0 ± 11.1, respectively.”

In addition, although model is designated as ANOVA under the table covariates are provided (?). There was a difference in the groups for additional variables (eg age)  and doing ANCOVA would be justified.

RESPONSE: We added an explanation of ANCOVA in section 2.3. Analysis and Statistics of the Methods section as follows:

Controlling age, career length, and training time, the MPS and ESCM scores between the two groups were analyzed with ANCOVA tests. The effect size of partial eta-squared for ANCOVA was interpreted as follows: partial Å‹2 = 0.01–0.09, small; Å‹2 = 0.09–0.25, medium; Å‹2 > 0.25, large.

To examine the relationship between career length and MPS in both professional and amateur golfers, partial correlations, controlling for age and training time, were used. To examine the relationship between handicaps and MPS in both professional and amateur golfers, partial correlations, controlling for age, career length, and training time, were used.

Controlling for age, career length, and training time, partial correlations were used to examine the relationship between MPS and ESCM in both professional and amateur golfers. The correlation coefficient effect size (Pearson’s r) was interpreted as follows: 0.1 < r < 0.2, small; 0.3 < r < 0.5, moderate; and r > 0.6, large. Statistical significance was set at p ≤ 0.05.”

Results_ Table 1 shows that the authors have introduced covariates: age, carrier years, training time. Nowhere else has it been written that ANCOVA was performed.

RESPONSE: We revised the statistical methods (section 2.3. Analysis and Statistics) and the legend of Table 1 to include this information.

In the correlation analyzes, the relationship between perfectionism and career years was presented first, while the main goal was asses of relationship between perfectionism and commitment as well as perfectionism and handicap. The reader has not been ealier prepared for presented the relationship between perfectionism and carrier years.

RESPONSE: We revised the presentation of the correlations as follows:

“Professional golfers’ MPS total scores were negatively associated with commitment-Cognition scores (r = -0.31, p < 0.01, -0.388 < 95% confidence interval [CI] < -0.228, moderate ES), while amateur golfers did not show a correlation between MPS scores and commitment-Cognition (r = 0.02, p = 0.82, -0.108 < 95% CI < 0.1485, small ES). In both professional (r = 0.12, p = 0.17, 0.032 < 95% CI < 0.206, small ES) and amateur golfers (r = -0.13, p < 0.01, -0.216 < 95% CI < -0.042, small ES), commitment-Cognition scores were not associated with handicap.

In both professional golfers (r = -0.40, p < 0.01, -0.472 < 95% CI < -0.323, moderate ES) and amateur golfers (r = -0.56, p < 0.01, -0.642 < 95% CI < -0.465, large ES), career length was negatively correlated with handicap. In professional golfers, handicaps were positively correlated with MPS total scores (r = 0.36, p < 0.01, 0.281 < 95% CI < 0.435, moderate ES), MPS other-oriented scores (r = 0.36, p < 0.01, 0.281 < 95% CI < 0.435, moderate ES), and MPS socially prescribed scores (r = 0.39, p < 0.01, 0.312 < 95% CI < 0.462, moderate ES). Amateur golfers did not show a correlation between MPS total scores and handicaps (r = 0.13, p = 0.15, 0.002 < 95% CI < 0.253, small ES).”

The authors write "In both professional (r = -0.40, p <0.001) and amateur golfers (r = 0.12, p = 0.17), commitment-Cognition scores were not associated with handicap (r = -0.13, p <0.001)" The content of the sentence is not adequate to the given values of p.

The authors present correlations for two groups. To compare them, you need to test for the significance of differences for correlation coefficients in two different groups.

RESPONSE: We corrected the r value in the text and table.

“In both professional (r = 0.12, p = 0.17, 0.032 < 95% CI < 0.206, small ES) and amateur golfers (r = -0.13, p < 0.01, -0.216 < 95% CI < -0.042, small ES), commitment-Cognition scores were not associated with handicap.”

Discussion

 In the discussion, the authors should refer more to the results. The authors should try to interpret the differences / similarities obtained in comparisons two groups. Eg why the correlations between MPS and Commitment differ between professionals and amateurs.

RESPONSE: We added the interpretation of why the correlation between MPS and commitment differs between professionals and amateurs in the Discussion.

In a comparison of professional and amateur golf players, perfectionism scores (overall and as related to oneself and to others) were found to be higher in the amateur golfers. These differences supported the theory that non-professional athletes would be associated with higher levels of perfectionistic strivings in sport. These results are especially interesting in light of the findings of Kang et al. [33] who reported that perfectionistic athletes felt threatened, which resulted in an increase in anxiety and the perception that evaluative situations were opportunities for failure. High perfectionists were also found to incur more stress and experience greater depression than their less perfectionist peers [34].

We also found that commitment scores were higher in professional golfers than in amateur golfers. Interestingly, both commitment subscales (Cognition and Behavior) were significantly different between professional and amateur golfers. The results of this study are consistent with Kang et al. [33] who reported that a group of individuals in the top 10% of the KLPGA had higher commitment scores than those in a non-KLPGA group. These findings suggest that high level elite athlete performance is moderately positively associated with commitment intensity. Similarly, these findings suggest that elite golfers perceived commitment to be at least potentially controllable, as has been the case with previous research [35,36].

Pearson’s correlation results supported our hypothesis that handicap in professional golfers was positively correlated with total scores for perfectionism, other-oriented, and socially prescribed perfectionism. In general psychology literature, people who scored highly in perfectionism were more likely to report negative thoughts across their life span [37,38]. In addition, athletes who scored highly in other-oriented and socially prescribed perfectionism were positively correlated with maladaptive outcomes in sports [4]. Prior research on perfectionism has identified both positive and negative aspects to this personality trait [39,40]. In particular, negative perfectionists are overly self-critical, rarely feel competent in carrying out their responsibilities and duties, and consistently doubt the quality of their performance [41]. This is because high obsessive traits have been found to increase the likelihood of perfectionist traits involving anxiety and distress [42].

In professional golfers, perfectionism scores were negatively correlated with the commitment-Cognition scores. In general psychology literature, perfectionist traits are closely associated with overcommitment [43]. Such a finding suggests that faster processing speeds involved in working memory capacity are positively correlated with a high commitment [24].

Although simple skills can be learned through repetitive practices, learning complex skills, such as developing one’s senses and awareness, can be more effective through individual (independent) sports [44]. Commitment plays an important role in one’s successful performance in sports by increasing intellectual capacity, allowing for accurate decisions to be made and consistent prediction in different situations [45]. In this context, elite golfers are more likely to have high commitment when planning strategies and having detailed plans during competitions.

It was hypothesized that perfectionist traits in professional golfers would negatively correlate with levels of commitment, but positively correlate with handicap. Based on the results of the current study, high-level performance in golfers is marked by lower perfectionism and high commitment during competition. These findings would suggest psychological training and performance enhancement in golfers. Controlling obsessive perfectionism in golfers may encourage the desire to participate in the sport, which can lead to improved performance reflected by an improvement in their handicap.”

Reviewer 4 Report

This manuscript (ms.) presents important theoretical and methodological/statistical limitations that allow me to conclude that ms. cannot be published in its current form. Reasons:

ABSTRACT:

Authors should explicitly indicate the type of sampling performed on both samples.

Indicate % males or females in both samples.

Indicate the type of statistical analysis conducted.

INTRODUCTION

This section is unsuccessful because the authors review, fundamentally, studies ejecuted in Western countries. This is not possible and appropriate as previous empirical evidence has systematically revealed cross-cultural differences respect to the perfectionism construct. E.g.:

DiBartolo & Redón (2012). A critical camination of the construct of perfectionism a

Lee & Park (2011). Cros-cultural validity of the Frost Multidimensional Perfectionism Scale in Korea. The Counseling Psychologists,

Harprit & Jojanjist (2011). Perfectionism and procastination: Cross-cultural perspective. J. of Social Sciences.

Therefore, the new contribution/novelty/originality of this ms. is not appropriate.

This section is unsuccessful because: The authors review, fundamentally, studies carried out in Western countries. Therefore, this is not possible and appropriate as previous empirical evidence has systematically revealed cross-cultural differences with respect to the perfectionism construct. E.g.: sssssssssssss. Therefore the new contribution/novelty/originality of this ms. in Korean people is poor and not shown correctly.

STATISTICAL ANALYSIS

The authors do not describe (Results section) and interpret (Discussion section) the effect sizes (e.g.., Cohen, 1998) for each of the types of statistical analysis presented. This is very important as it invalidates the conclusions drawn in this ms..

Author Response

Reviewer 4

ABSTRACT:

Authors should explicitly indicate the type of sampling performed on both samples.

RESPONSE: We added the type of sampling as follows:

“Using simple random sampling, 486 professional golfers ”

Indicate % males or females in both samples.

RESPONSE: We added % females as follows:

486 professional golfers (mean age = 22.1 ± 3.0, 52.1% female) and 233 amateur golfers (mean age = 44.8 ± 10.2, 55.8% female) were recruited.”

Indicate the type of statistical analysis conducted.

RESPONSE: We added the type of statistical analysis conducted as follows:

“An ANCOVA, controlling for age, career length, and training time, revealed lower MPS self-oriented scores (-10.3%; F = 8.9, p < 0.01; Å‹2 = 0.498), and higher ESCM-Cognition (12.6%; F = 9.4, p < 0.01; Å‹2 = 0.691) and ESCM-Behavior (9.4%; F = 4.6, p = 0.03; Å‹2 = 0.479) scores in professional golfers than amateur golfers. In partial correlations controlling for age, career length, and training time, MPS scores were negatively associated with ESCM-Cognition scores (r = -0.31, p < 0.001) only in professional golfers. Professional golfers’ mean handicaps were positively correlated with MPS total (r = 0.36, p < 0.01).”

INTRODUCTION

This section is unsuccessful because the authors review, fundamentally, studies ejecuted in Western countries. This is not possible and appropriate as previous empirical evidence has systematically revealed cross-cultural differences respect to the perfectionism construct. E.g.:

DiBartolo & Redón (2012). A critical camination of the construct of perfectionism a

Lee & Park (2011). Cros-cultural validity of the Frost Multidimensional Perfectionism Scale in Korea. The Counseling Psychologists,

Harprit & Jojanjist (2011). Perfectionism and procastination: Cross-cultural perspective. J. of Social Sciences.

Therefore, the new contribution/novelty/originality of this ms. is not appropriate.

This section is unsuccessful because: The authors review, fundamentally, studies carried out in Western countries. Therefore, this is not possible and appropriate as previous empirical evidence has systematically revealed cross-cultural differences with respect to the perfectionism construct. E.g.: sssssssssssss. Therefore the new contribution/novelty/originality of this ms. in Korean people is poor and not shown correctly.

RESPONSE: We added several studies that examined perfectionism in Korean athletes in the Introduction section.

“There have been several studies about perfectionism in Korean athletes [2,3]. Yoon and Kim [2] reported that the traits of perfectionism in man-to-man sports were higher than those in record genre sports. Bum et al. [3] reported that self-oriented perfectionism and socially prescribed perfectionism would aggravate the level of stress and emotional exhaustion.”

  1. Yoon, K.; Kim, T. The relationship between perfectionism and motivational climate in competitive athletes. Digital Convergence 2019, 17, 369–376.
  2. Bum, C.H.; Yoo, C.K.; Jung, C.K. A convergence study on the relationship between perfectionism, stress, and burnout among college golf athletes. J. Korea Convergence Soc. 2017, 8, 243–252.

→ We also added your opinion about cultural differences to the Limitations section as follows:

Finally, perfectionism is thought to be affected by cultural differences. Therefore, readers should be cautious in generalizing the current results. Further studies should focus on assessing factors such as sports stress, psychological factors, physical injuries, and cultural differences in golfers.”

STATISTICAL ANALYSIS

The authors do not describe (Results section) and interpret (Discussion section) the effect sizes (e.g.., Cohen, 1998) for each of the types of statistical analysis presented. This is very important as it invalidates the conclusions drawn in this ms..

RESPONSE: We revised the statistical analysis section (section 2.3. Analysis and Statistics) and the Results section when presenting the statistical values. In addition, we revised the Discussion section when interpreting the results.

“Analysis and Statistics

     Statistical analysis was performed using commercial software (SPSS for Windows, ver. 18.0; SPSS Inc., Chicago, IL, USA). Using a Kolmogorov-Smirnov test, all data were tested for normality. Age, career length (years), training time, and handicap between the two groups were analyzed with independent t-tests. The effect size of Cohen’s d for independent t-tests was interpreted as follows: 0.0 < d < 0.2, small; 0.3 < d < 0.5, medium; and d > 0.6, large. Sex distribution between the two groups was analyzed with a Chi-squared test. The effect size of Cramer’s V for Chi-squared tests was interpreted as follows: V > 0, no or very weak; V > 0.05, weak; V > 0.10, moderate; V > 0.15, medium; and V > 0.25, very strong.

Controlling age, career length, and training time, the MPS and ESCM scores between the two groups were analyzed with ANCOVA tests. The effect size of partial eta-squared for ANCOVA was interpreted as follows: partial Å‹2 = 0.01–0.09, small; Å‹2 = 0.09–0.25, medium; and Å‹2 > 0.25, large.

To examine the relationship between career length and MPS in both professional and amateur golfers, partial correlations, controlling for age and training time, were used. To examine the relationship between handicaps and MPS in both professional and amateur golfers, partial correlations, controlling for age, career length, and training time, were used.

Controlling for age, career length, and training time, partial correlations were used to examine the relationship between MPS and ESCM in both professional and amateur golfers. The correlation coefficient effect size (Pearson’s r) was interpreted as follows: 0.1 < r < 0.2, small; 0.3 < r < 0.5, moderate; and r > 0.6, large. Statistical significance was set at p ≤ 0.05.

Demographic and Psychological Characteristics

All data including age (professional golfer: skewness z = 1.74, kurtosis z = 1.01; amateur golfer: skewness z = -1.10, kurtosis z = 1.39), career length (professional golfer: skewness z = 1.63, kurtosis z = 1.16; amateur golfer: skewness z = -1.03, kurtosis z = -1.01), training time (professional golfer: skewness z = 0.72, kurtosis z = 1.15; amateur golfer: skewness z = 0.78, kurtosis z = 0.65), handicap (professional golfer: skewness z = -0.67, kurtosis z = -0.53; amateur golfer: skewness z = 1.05, kurtosis z = 1.09), MPS total score (professional golfer: skewness z = -0.61, kurtosis z = -0.48; amateur golfer: skewness z =-1.15, kurtosis z = 1.02), commitment-Cognition score (professional golfer: skewness z = -1.24, kurtosis z = 1.01; amateur golfer: skewness z = 0.11, kurtosis z = -0.29), and commitment-Behavior score (professional golfer: skewness z = -0.33, kurtosis z = 0.36; amateur golfer: skewness z = 0.67, kurtosis z = -0.93) were slightly skewed and kurtotic for both professional and amateur golfers, but they did not differ significantly from normality.

There was no significant difference in the sex distribution between the two groups (χ2 = 0.8, p = 0.34, effect size [ES] = 0.035, small ES). Professional golfers were younger (-102.7%, t = 45.9, p < 0.01, ES = 3.012, large ES), and had fewer career years (-55.6%, t = 8.7, p < 0.01, ES = 0.592, medium ES) and lower handicaps (-13.9%, t = 15.1, p < 0.01, ES = 1.308, large ES), compared to amateur golfers. Professional golfers showed longer training time, compared to amateur golfers (50.7%, t = 13.3, p < 0.01, ES = 1.293).

Professional golfers had lower MPS total scores (-4.7%, F = 4.8, p = 0.02, ES = 0.341, large ES), as well as lower MPS self-oriented scores (-10.3%; F = 8.9, p < 0.01, ES = 0.498, large ES), compared to amateur golfers. Also, professional golfers showed lower MPS other-oriented scores at a trend level compared to amateur golfers (-1.2%, F = 3.7, p = 0.06, ES = 0.224, moderate ES). Finally, professional golfers showed higher commitment-Cognition scores (12.6%, F = 9.4, p < 0.01, ES = 0.691, large ES) and commitment-Behavior scores (9.4%, F = 4.6, p = 0.03, ES = 0.479, large ES) compared to amateur golfers.

Correlation Between Perfectionism, Commitment, Handicap, and Career Length

Professional golfers’ MPS total scores were negatively associated with commitment-Cognition scores (r = -0.31, p < 0.01, -0.388 < 95% confidence interval [CI] < -0.228, moderate ES), while amateur golfers did not show a correlation between MPS scores and commitment-Cognition (r = 0.02, p = 0.82, -0.108 < 95% CI < 0.1485, small ES). In both professional (r = 0.12, p = 0.17, 0.032 < 95% CI < 0.206, small ES) and amateur golfers (r = -0.13, p < 0.01, -0.216 < 95% CI < -0.042, small ES), commitment-Cognition scores were not associated with handicap.

In both professional golfers (r = -0.40, p < 0.01, -0.472 < 95% CI < -0.323, moderate ES) and amateur golfers (r = -0.56, p < 0.01, -0.642 < 95% CI < -0.465, large ES), career length was negatively correlated with handicap. In professional golfers, handicaps were positively correlated with MPS total scores (r = 0.36, p < 0.01, 0.281 < 95% CI < 0.435, moderate ES), MPS other-oriented scores (r = 0.36, p < 0.01, 0.281 < 95% CI < 0.435, moderate ES), and MPS socially prescribed scores (r = 0.39, p < 0.01, 0.312 < 95% CI < 0.462, moderate ES). Amateur golfers did not show a correlation between MPS total scores and handicaps (r = 0.13, p = 0.15, 0.002 < 95% CI < 0.253, small ES).”

In a comparison of professional and amateur golf players, perfectionism scores (overall and as related to oneself and to others) were found to be higher in the amateur golfers. These differences supported the theory that non-professional athletes would be associated with higher levels of perfectionistic strivings in sport. These results are especially interesting in light of the findings of Kang et al. [33] who reported that perfectionistic athletes felt threatened, which resulted in an increase in anxiety and the perception that evaluative situations were opportunities for failure. High perfectionists were also found to incur more stress and experience greater depression than their less perfectionist peers [34].

We also found that commitment scores were higher in professional golfers than in amateur golfers. Interestingly, both commitment subscales (Cognition and Behavior) were significantly different between professional and amateur golfers. The results of this study are consistent with Kang et al. [33] who reported that a group of individuals in the top 10% of the KLPGA had higher commitment scores than those in a non-KLPGA group. These findings suggest that high level elite athlete performance is moderately positively associated with commitment intensity. Similarly, these findings suggest that elite golfers perceived commitment to be at least potentially controllable, as has been the case with previous research [35,36].

Pearson’s correlation results supported our hypothesis that handicap in professional golfers was positively correlated with total scores for perfectionism, other-oriented, and socially prescribed perfectionism. In general psychology literature, people who scored highly in perfectionism were more likely to report negative thoughts across their life span [37,38]. In addition, athletes who scored highly in other-oriented and socially prescribed perfectionism were positively correlated with maladaptive outcomes in sports [4]. Prior research on perfectionism has identified both positive and negative aspects to this personality trait [39,40]. In particular, negative perfectionists are overly self-critical, rarely feel competent in carrying out their responsibilities and duties, and consistently doubt the quality of their performance [41]. This is because high obsessive traits have been found to increase the likelihood of perfectionist traits involving anxiety and distress [42].

In professional golfers, perfectionism scores were negatively correlated with the commitment-Cognition scores. In general psychology literature, perfectionist traits are closely associated with overcommitment [43]. Such a finding suggests that faster processing speeds involved in working memory capacity are positively correlated with a high commitment [24].

Although simple skills can be learned through repetitive practices, learning complex skills, such as developing one’s senses and awareness, can be more effective through individual (independent) sports [44]. Commitment plays an important role in one’s successful performance in sports by increasing intellectual capacity, allowing for accurate decisions to be made and consistent prediction in different situations [45]. In this context, elite golfers are more likely to have high commitment when planning strategies and having detailed plans during competitions.

It was hypothesized that perfectionist traits in professional golfers would negatively correlate with levels of commitment, but positively correlate with handicap. Based on the results of the current study, high-level performance in golfers is marked by lower perfectionism and high commitment during competition. These findings would suggest psychological training and performance enhancement in golfers. Controlling obsessive perfectionism in golfers may encourage the desire to participate in the sport, which can lead to improved performance reflected by an improvement in their handicap.”

Round 2

Reviewer 2 Report

The authors made a significant improvement to the article. My suggestions were considered during the process. I would like to endorse acceptance.

Author Response

Thank you for your comments.

Reviewer 3 Report

The authors have made many changes. I appreciate their great effort. However, in the discription of statistical analysis and research results, unfortunately new, quite big problems appeared.

The description of the analysis in this version of the paper shows that the authors changed the classic Pearson correlations (previous version) for partial correlations (actual version). However, in many cases, the correlations that the authors gave in this version of the paper have the same values as those given in the previous version of the paper (correlations between MPS total scores and Commitmen-Cognition line: 175-178, career length and hendicap line: 181- 183, correlations between handicaps and MPS line: 183-187). It is difficult to assume  that in these cases the values of partial correlations (when controlling career length, age, training time) will be exactly the same as the values of Pearson's coefficients (zero order correlation).

Also the description under the figure suggests that these are partial correlations. However, the figures  shows the relationships between variables in their original scales (as linear regressions). This solution can eventually addopted for Pearson correlations, but  this is not adequate for partial correlations.

I suggest that the authors should seek professional statistical consultation.

Author Response

The authors have made many changes. I appreciate their great effort. However, in the discription of statistical analysis and research results, unfortunately new, quite big problems appeared.

The description of the analysis in this version of the paper shows that the authors changed the classic Pearson correlations (previous version) for partial correlations (actual version). However, in many cases, the correlations that the authors gave in this version of the paper have the same values as those given in the previous version of the paper (correlations between MPS total scores and Commitmen-Cognition line: 175-178, career length and hendicap line: 181- 183, correlations between handicaps and MPS line: 183-187). It is difficult to assume that in these cases the values of partial correlations (when controlling career length, age, training time) will be exactly the same as the values of Pearson's coefficients (zero order correlation).

→ We are sorry that we missed the correction of r and p values in result section of text. We revise it in result section.

Also the description under the figure suggests that these are partial correlations. However, the figures shows the relationships between variables in their original scales (as linear regressions). This solution can eventually addopted for Pearson correlations, but this is not adequate for partial correlations.

I suggest that the authors should seek professional statistical consultation.

→ We removed ‘linear fit line’ from figures and just showed scatter plots.